# Wireworm (Coleoptera: Elateridae) genomic analysis reveals putative cryptic species, population structure, and adaptation to pest control

Kimberly R. Andrews [1✉], Alida Gerritsen[2], Arash Rashed [3], David W. Crowder[4], Silvia I. Rondon [5], Willem G. van Herk[6], Robert Vernon[7], Kevin W. Wanner[8], Cathy M. Wilson[9], Daniel D. New[1], Matthew W. Fagnan[1], Paul A. Hohenlohe[1] & Samuel S. Hunter [1]

The larvae of click beetles (Coleoptera: Elateridae), known as "wireworms," are agricultural pests that pose a substantial economic threat worldwide. We produced one of the first wireworm genome assemblies (*Limonius californicus*), and investigated population structure and phylogenetic relationships of three species (*L. californicus, L. infuscatus, L. canus*) across the northwest US and southwest Canada using genome-wide markers (RADseq) and genome skimming. We found two species (*L. californicus* and *L. infuscatus*) are comprised of multiple genetically distinct groups that diverged in the Pleistocene but have no known distinguishing morphological characters, and therefore could be considered cryptic species complexes. We also found within-species population structure across relatively short geographic distances. Genome scans for selection provided preliminary evidence for signatures of adaptation associated with different pesticide treatments in an agricultural field trial for *L. canus*. We demonstrate that genomic tools can be a strong asset in developing effective wireworm control strategies.

[1] Institute for Bioinformatics and Evolutionary Studies (IBEST), University of Idaho, Moscow, ID 83844, USA. [2] Computational Sciences Center, National Renewable Energy Laboratory, Golden, CO 80401, USA. [3] Department of Entomology, Plant Pathology and Nematology, University of Idaho, Moscow, ID 83844, USA. [4] Department of Entomology, Washington State University, Pullman, WA 99164, USA. [5] Oregon State University, Hermiston Agricultural Research and Extension Center, Hermiston, OR 97838, USA. [6] Agassiz Research and Development Centre, Agriculture and Agri-Food Canada, Agassiz, British Columbia, Canada V0M 1A0. [7] Sentinel IPM Services, Chilliwack, British Columbia, Canada V2R 3B5. [8] Department of Plant Sciences and Plant Pathology, Montana State University, Bozeman, MT 59717, USA. [9] Idaho Wheat Commission, Boise, ID 83702, USA. ✉email: kimberlya@uidaho.edu

The larvae of click beetles (Coleoptera: Elateridae), known as "wireworms", pose a growing threat to agricultural crops in temperate and subtropical regions around the world[1,2]. Wireworms feed on the sprouting seedlings, root, and stem tissues of a wide range of economically important crops such as wheat (*Triticum* spp.), corn (*Zea mays* L.), and potatoes (*Solanum tuberosum* L.). They typically live in the soil for 2–11 years, depending on the species and environmental conditions. Wireworms are becoming increasingly prevalent in croplands because pesticides that effectively controlled them are no longer commercially available due to concerns regarding human and environmental health[3,4]. Current methods for controlling wireworms are far less effective; the new generation insecticides are neonicotinoids, which only temporarily debilitate or repel the wireworms, without causing substantial mortality[5–10]. Moreover, alternative strategies involving cultural practices such as crop rotation are not highly effective and are not feasible in many cropping systems, primarily due to the wide host range of this pest complex[1,2].

The lack of effective methods to control wireworm infestations has led to a growing interest in developing integrated pest management (IPM) strategies for these species[2,11]. IPM utilizes a combination of biological, cultural, mechanical, and chemical controls, guided by knowledge of the biology and ecology of the pest species[12,13]. This approach relies on the idea that different pest species, or different populations of pest species, may respond differently to control measures; therefore, one of the first steps for effective IPM is to determine which species and populations are present in a given area. However, knowledge of wireworm species composition and population structure is often limited, thereby reducing the ability to design and implement IPM strategies. One reason for this lack of information is that wireworm species in the larval stage are difficult to identify morphologically due to a lack of clear distinguishing characters between species; only a limited number of experts in the field are able to distinguish some species. This limitation is exacerbated by the scarcely characterized population structure of wireworms.

Molecular approaches have strong potential to fill knowledge gaps and address practical challenges regarding wireworm species identification and population structure. DNA barcoding is one example of a molecular tool that relies on sequence data from mitochondrial DNA (mtDNA) markers to distinguish species[14]. This technique can be particularly useful when morphology-based species identification is difficult. DNA barcoding efforts on wireworms from North America, Europe, and other regions have demonstrated that portions of the *cytochrome oxidase I* (*COI*) gene and *16S ribosomal DNA* (rDNA) are effective mtDNA markers for distinguishing known species[15–19]. These studies also found genetic evidence for the presence of wireworm species that were previously undescribed due to the absence of distinct morphological traits between them (cryptic species), indicating that the number of wireworm species may currently be underestimated.

Molecular approaches also offer effective methods in investigating within-species population structure and dispersal patterns. Genetic analyses investigating population structure ideally use multiple nuclear markers. In the past, population genetic studies of non-model organisms typically used 10–20 nuclear microsatellite markers, but new high-throughput DNA sequencing technologies have led to the increasing adoption of higher-resolution approaches using thousands of single nucleotide polymorphisms (SNPs) scattered across the nuclear genome[20]. Data from thousands of SNPs also enables genome scan approaches to identify genomic regions under different adaptive pressures in different habitats[21]. These approaches can be used to assess whether populations are locally adapted at the genomic level to different habitats, information that could help predict how populations will respond to control measures. In addition, genome scan approaches can be used to evaluate whether populations are evolving over time in response to environmental change, such as changes in pesticide treatment or IPM strategies; this evolution could lead to a reduction in the efficacy of treatment over time. However, despite the strong potential utility of genetic approaches for the development of effective IPM for wireworms, to our knowledge no population genetic or genome scan approaches have been conducted for wireworm species.

Here, we use genomic techniques to investigate putative cryptic species, within-species population genetic structure, and local adaptation of wireworms across a range of geographic scales in the northwest US and southwest Canada. We focus on three wireworm species that are resurging as pests: *Limonius californicus* Mannerheim, *L. infuscatus* Motschulsky, and *L. canus* LeConte (Fig. 1)[22,23]. These species are endemic to the northwest US and southwest Canada, with *L. californicus* also occurring throughout California[24]. Previous DNA barcoding studies found evidence for putative cryptic species within both *L. californicus* and *L. infuscatus* in this region[17,18]. We build on these studies by expanding the geographic range of sampling, and by using high-resolution genome-wide sequence data from both nuclear and mtDNA markers. We sequenced and assembled the whole genome for one *L. californicus* specimen, and generated genomic data from multiple individuals from each of the three species using restriction site-associated DNA sequencing (RADseq), a technique that surveys thousands of loci across the nuclear genome[25]. We also generated mtDNA sequence data from multiple individuals using genome skimming, a technique that uses low-coverage shotgun sequencing to retrieve mtDNA sequence data[26,27]. We identified genetic lineages within both *L. californicus* and *L. infuscatus* that were highly divergent for both RADseq and mtDNA markers, potentially indicating the presence of cryptic species complexes. We also found evidence for fine-scale within-species population structure for the *L. californicus* and *L. infuscatus* species complexes, with genetically divergent populations separated by as little as 25 km. We did not find genetically distinct groups within a single agricultural field for *L. canus*, but genome scans for selection indicated that pesticide treatments may be driving adaptive evolution within the agricultural field. The results of our study provide insight into the species composition, population genetic structure, and genomic adaptation of wireworms in the northwest US and southwest Canada that can assist in the development of effective IPM programs.

## Results

**Shotgun sequencing and genome assembly**. Shotgun sequencing using genomic DNA from a single *L. californicus* specimen produced 160 million PE250 reads (80 Gbp) from Illumina HiSeq 2500, 33 million PE300 reads (19.9 Gbp) from Illumina MiSeq, 738,071 synthetic long reads with median length 1253 bp (1.47 Gbp) from Illumina SLR, and 1,750,383 reads with median length 2400 bp from PacBio RSII (5.01 Gbp). Assembly produced 115,615 contigs with an assembled length of 1,072,695,639 bp, and N50 of 19,399 bp. BUSCO analysis reported that 1476 of the 1658 reference genes were complete (89%, 1387 single copy and 89 duplicated), 137 (8.3%) were fragmented, and 45 (2.7%) were missing in the draft assembly, indicating that the assembly represents most of the genomic content.

**RADseq filtering**. Prior to generating RADseq data, each wireworm sample was assigned a species designation based on visual morphological traits assessed under a dissecting microscope

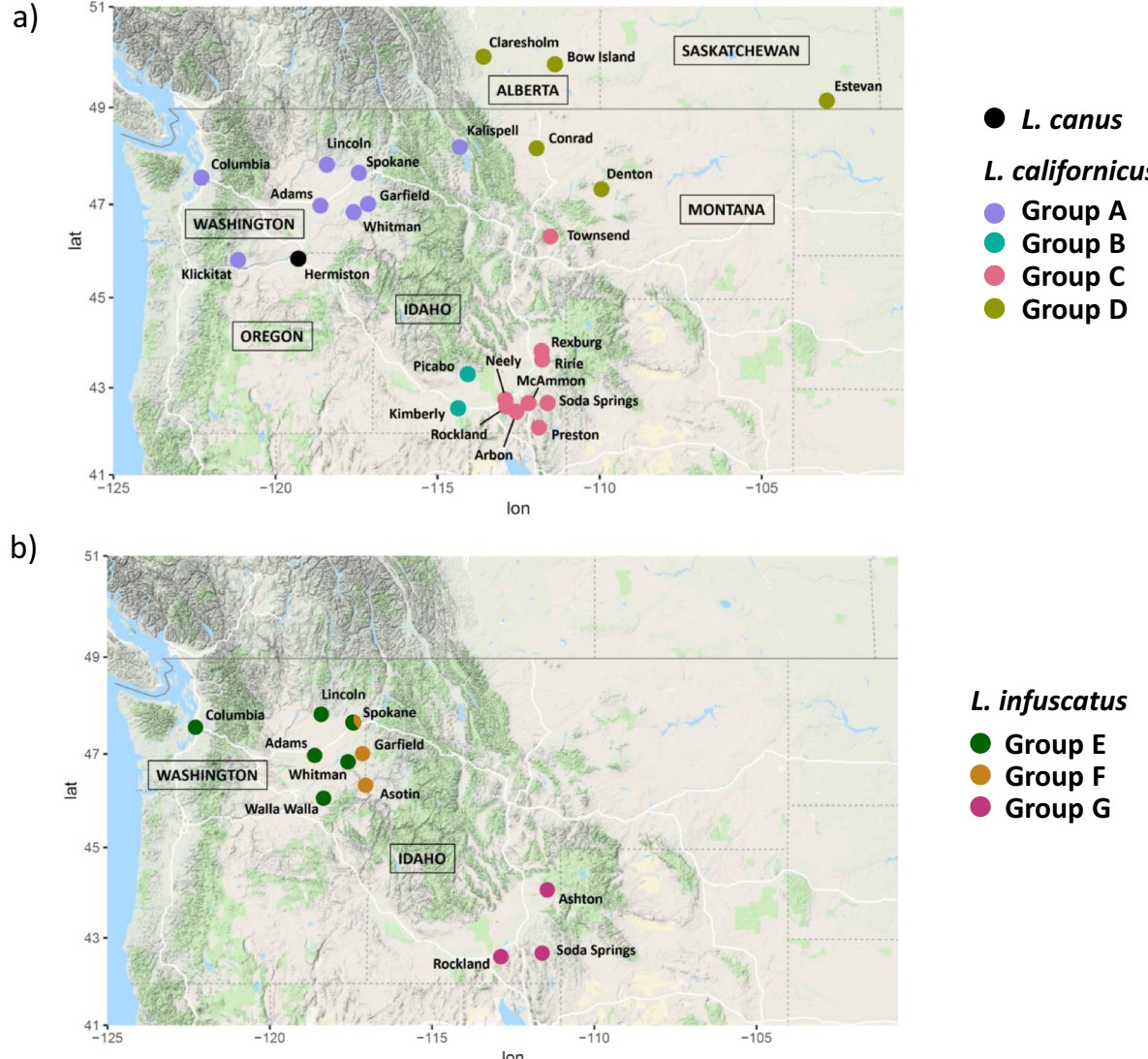

**Fig. 1 Map of sampling locations color-coded by genetically distinct groups identified using PCA and sNMF. a** *L. canus*, colored black; and *L. californicus*, with group names (A–D) following the circles in Fig. 4a. **b** *L. infuscatus*, with group names (E–G) following the circles in Fig. 4b. Samples from both Groups E and F were found in Spokane, and the pie chart shows the proportion of individuals from each group for that location. Map data: Google.

following the protocols of ref. [22] and ref. [23] (Fig. 1, Table 1, Supplementary Data 1). After filtering the RADseq data, a total of 247 wireworm samples were retained that had been collected from agricultural fields across the northwest US and southwest Canada. The mean number of reads per sample was 1,041,596.2 (range 394,488–3,300,186, with forward and reverse reads counted separately). The average mapping rate was 96.6% (range 76.3–98.7%) for *L. californicus* samples, 72.8% (range 63.0–76.6%) for *L. infuscatus* samples, and 94.2% (range 83.4–95.9%) for *L. canus* samples. After filtering, the number of SNPs retained for the full dataset with all three species was 3770. The number of SNPs retained for analyses conducted separately for each species was 2189 for *L. californicus*, 584 for *L. infuscatus,* and 13,725 for *L. canus*. The number of SNPs retained for analyses conducted separately for each *L. californicus* subgroup (described further below) ranged from 489 to 1118, and for each *L. infuscatus* subgroup ranged from 223 to 330. The number of SNPs was relatively low for *L. californicus* because we did not allow missing data for the *L. californicus* analyses during filtering (see the "Methods" section). The number of SNPs was lowest for the *L. infuscatus* analyses due to a lower rate of mapping to the *L. californicus* genome, indicative

of a relatively high evolutionary divergence between this species and *L. californicus*.

**Genetic structure: between known species**. Principal components analysis (PCA) explained a high proportion of genetic variation (61.1% and 24.7% for the first and second axes, respectively) and separated the three recognized *Limonius* species, with *L. infuscatus* separating on the first axis, and the other two species separating on the second axis (Fig. 2). This PCA was conducted after subsampling each species to account for uneven sample sizes, which can strongly influence PCA[28]. When including all samples in the PCA, the three species still separated, but *L. canus* separated on the first axis due to a much larger sample size (Supplementary Fig. 1). Two exceptions were observed in which samples did not cluster according to their morphologically identified species: one *L. californicus* sample from Ashton, Idaho grouped with *L. infuscatus*, and one *L. californicus* sample from Garfield, Washington grouped with *L. canus*. Given the strong morphological similarity of many wireworm species, these samples were likely mis-identified

**Table 1 Sample sizes across locations included in RADseq analysis for *L. californicus*, *L. canus*, and *L. infuscatus*.**

| State or province | City or county | Pesticide treatment | *L. californicus* | *L. canus* | *L. infuscatus* |
|---|---|---|---|---|---|
| Alberta | Bow Island | NA | 4 | | |
| Alberta | Claresholm | NA | 4 | | |
| Saskatchewan | Estevan | NA | 3 | | |
| Idaho | Arbon | NA | 4 | | |
| Idaho | Ashton | NA | | | 3 |
| Idaho | Kimberly | NA | 3 | | |
| Idaho | McAmmon | NA | 2 | | |
| Idaho | Neeley | NA | 3 | | |
| Idaho | Picabo | NA | 3 | | |
| Idaho | Preston | NA | 1 | | |
| Idaho | Rexburg | NA | 7 | | |
| Idaho | Ririe | NA | 6 | | |
| Idaho | Rockland | NA | 1 | | 2 |
| Idaho | Soda Springs | NA | 5 | | 1 |
| Idaho | Unknown | NA | 3 | | 2 |
| Montana | Conrad | NA | 1 | | |
| Montana | Denton | NA | 1 | | |
| Montana | Kalispell | NA | 1 | | |
| Montana | Townsend | NA | 1 | | |
| Washington | Adams | NA | 1 | | 1 |
| Washington | Asotin | NA | | | 1 |
| Washington | Columbia | NA | 9 | | 1 |
| Washington | Garfield | NA | 2 | 1 | 1 |
| Washington | Klickitat | NA | 2 | | |
| Washington | Lincoln | NA | 3 | | 6 |
| Washington | Spokane | NA | 3 | | 5 |
| Washington | Walla Walla | NA | | | 2 |
| Washington | Whitman | NA | 2 | | 3 |
| Oregon | Hermiston | Conventional | | 37 | |
| | | Semi-conventional | | 78 | |
| | | Semi-organic | | 28 | |
| Total | | | 75 | 144 | 28 |

Pesticide treatment type is indicated for *L. canus* samples from Hermiston, Oregon.

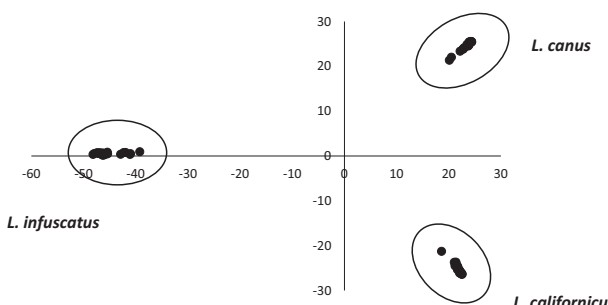

**Fig. 2 PCA of the RADseq dataset subsampled to equal sample sizes across the three recognized species.** The first axis accounted for 61.1% of the genetic variation, and the second axis accounted for 24.7% of the genetic variation.

morphologically. For 13 samples, the species identification had been unknown based on morphological data; of these, nine grouped with *L. californicus* and four grouped with *L. infuscatus*.

Sparse non-negative matrix factorization algorithms (sNMF) analysis of the full dataset also separated the three species (Fig. 3a, Supplementary Fig. 2a). Pairwise $F_{ST}$ values were high and strongly significant ($p < 0.00001$) between all three species, and were highest for comparisons including *L. infuscatus*: 0.816 for *L. californicus* vs. *L. canus*, 0.891 for *L. californicus* vs. *L. infuscatus*, and 0.904 for *L. canus* vs. *L. infuscatus*. Pairwise genetic distances followed a similar trend: 0.236 for *L. californicus* vs. *L. canus*, 0.596 for *L. californicus* vs. *L. infuscatus*, and 0.621 for *L. canus* vs. *L. infuscatus*.

**Genetic structure: within known species**. PCA and sNMF analyses were conducted independently for each species to investigate population structure. For *L. californicus*, the first and second axes of the PCA accounted for 26.9% and 20.7% of the genetic variation, respectively, and separated the samples into four main groups: (A) Washington and northwest Montana, (B) south-central Idaho, (C) southeast Idaho and southwest Montana, and (D) Canada and north central Montana (Fig. 4a). sNMF confirmed these four main groupings, with the best $K = 4$ (Supplementary Fig. 2b) and individuals separating into the same groups as the PCA (Fig. 3b). There were two exceptions to this grouping pattern for the PCA and sNMF analyses: First, PCA indicated that one of the nine samples from Columbia, Washington plotted between Groups A and B, and sNMF analysis also indicated this individual had a lower Group A ancestry and higher Group B ancestry than the other Group A samples (58.8% Group A, 26.9% Group B, 8.43% Group C, 5.96% Group D). Second, PCA indicated that the sample from southwest Montana (Townsend) plotted outside the main cluster of Idaho samples for Group C, and sNMF analysis also indicated that this individual had a lower Group C ancestry than the rest of the Group C samples (82.4% Group C, 17.6% Group D). Pairwise $F_{ST}$ values between all four groups were significant ($p < 0.001$) and ranged from 0.251 to 0.491 (Table 2).

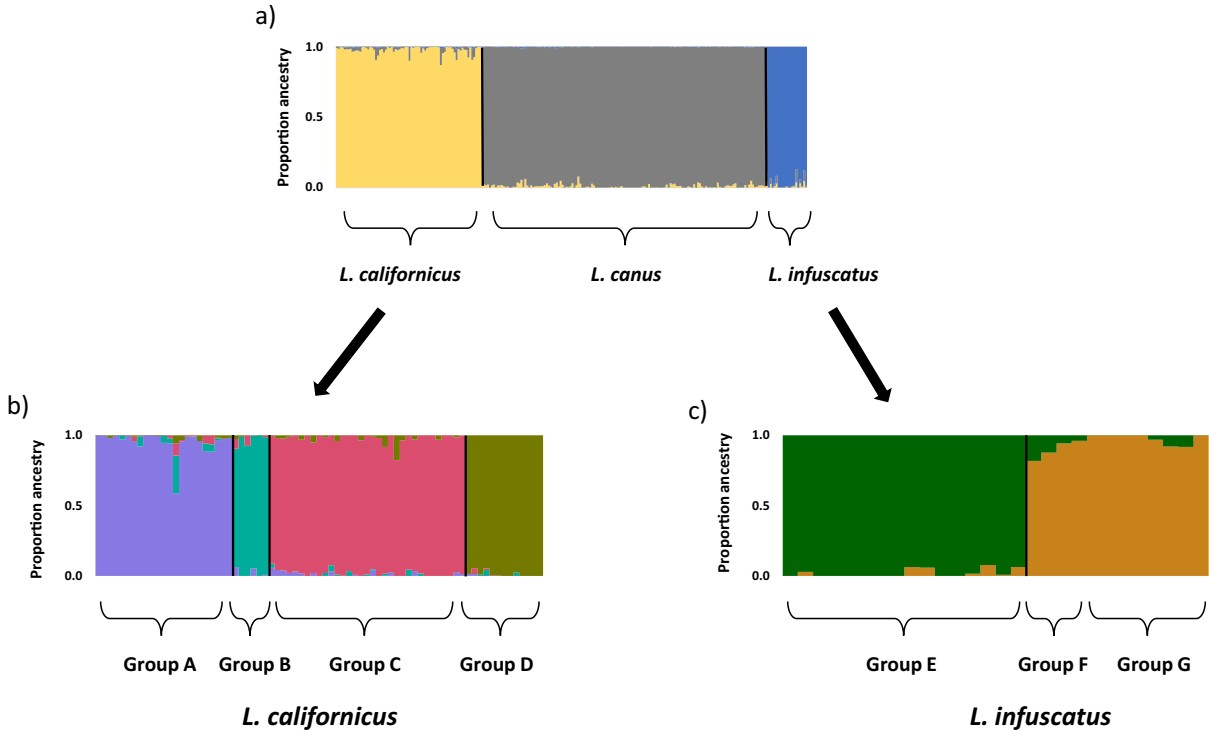

**Fig. 3 Ancestry proportions calculated using sNMF with RADseq data for the three recognized wireworm species.** Results are reported for the best *K* for **a** the full dataset, *K* = 3; **b** *L. californicus*, *K* = 4; and **c** *L. infuscatus*, *K* = 2. Group names and colors follow Fig. 1.

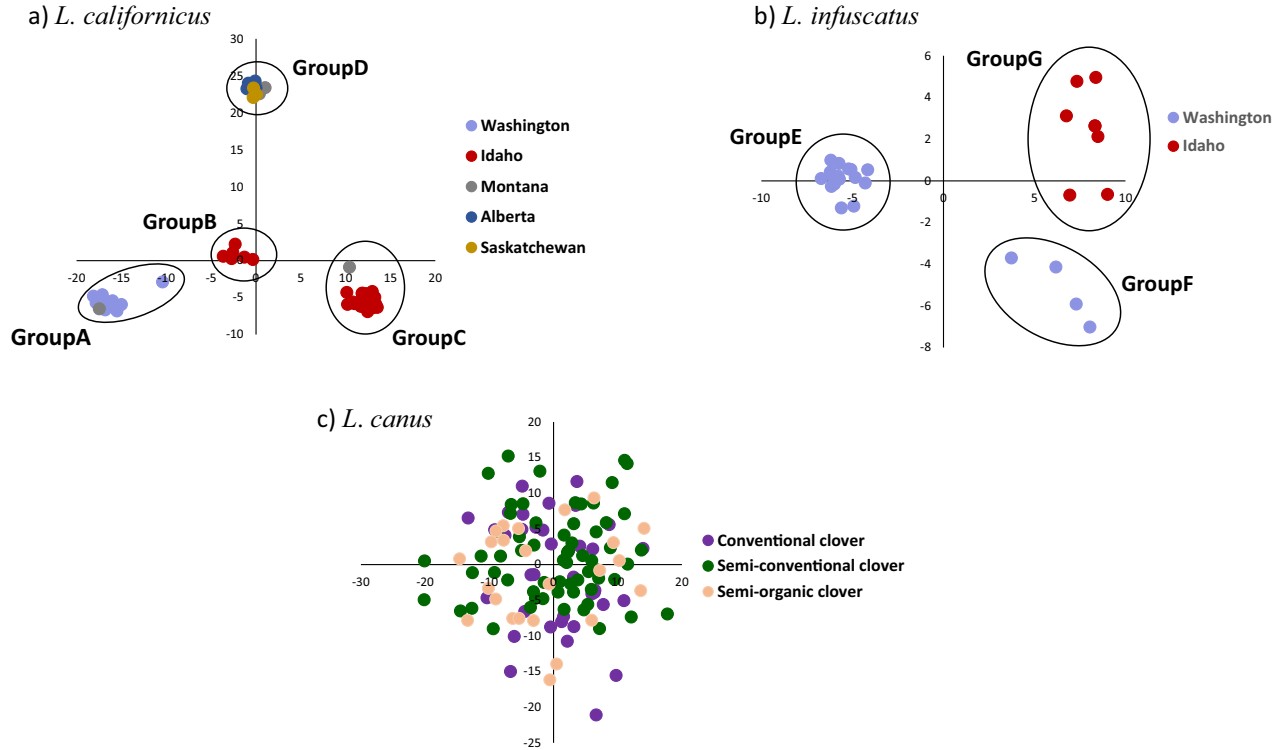

**Fig. 4 PCAs of the RADseq data for each of the three recognized wireworm species. a** *L. californicus*: first and second axes account for 26.9 and 20.7% of genetic variation, **b** *L. infuscatus*: first and second axes account for 37.7 and 6.31% of genetic variation, and **c** Hermiston, Oregon *L. canus*: first and second axes account for 2.33 and 1.84% of genetic variation.

**Table 2 Pairwise $F_{ST}$ for genetically distinct groups identified by PCA and sNMF for *L. californicus* and *L. infuscatus*, and across pesticide treatment plots for *L. canus*.**

| | | L. californicus | | | | L. canus | | | L. infuscatus | | |
|---|---|---|---|---|---|---|---|---|---|---|---|
| | | Group A | Group B | Group C | Group D | Conv | Semi-conv | Semi-org | Group E | Group F | Group G |
| *L. californicus* | Group A | | **0.000** | **0.000** | **0.000** | **0.000** | **0.000** | **0.000** | **0.000** | **0.000** | **0.000** |
| | Group B | **0.247** | | **0.000** | **0.000** | **0.000** | **0.000** | **0.000** | **0.000** | **0.011** | **0.000** |
| | Group C | **0.382** | **0.257** | | **0.000** | **0.000** | **0.000** | **0.000** | **0.000** | **0.000** | **0.000** |
| | Group D | **0.482** | **0.427** | **0.433** | | **0.000** | **0.000** | **0.000** | **0.000** | **0.001** | **0.000** |
| *L. canus* | Conv | **0.836** | **0.822** | **0.838** | **0.847** | | 0.083 | 0.315 | **0.000** | **0.000** | **0.000** |
| | Semi-conv | **0.832** | **0.823** | **0.833** | **0.841** | 0.000 | | 0.113 | **0.000** | **0.000** | **0.000** |
| | Semi-organic | **0.839** | **0.824** | **0.840** | **0.853** | −0.001 | 0.000 | | **0.000** | **0.001** | **0.000** |
| *L. infuscatus* | Group E | **0.922** | **0.917** | **0.920** | **0.936** | **0.909** | **0.908** | **0.911** | | **0.002** | **0.000** |
| | Group F | **0.915** | **0.904** | **0.914** | **0.939** | **0.898** | **0.899** | **0.900** | **0.304** | | 0.439 |
| | Group G | **0.915** | **0.900** | **0.916** | **0.930** | **0.904** | **0.906** | **0.905** | **0.335** | −0.035 | |

Group names follow Fig. 3. $F_{ST}$ values are below the diagonal, and $p$ values are above the diagonal. Bold values indicate $p < 0.05$.

For *L. infuscatus*, the first PCA axis accounted for 37.7% of the genetic variation and separated a group of samples collected from throughout Washington (Group E) from a group of both southeast Idaho and eastern Washington samples (Groups F and G) (Fig. 4b). Both of these groups included samples from Spokane, Washington, indicating the presence of two sympatric, genetically distinct populations in the Spokane area. The second axis accounted for less genetic variation (6.31%) and separated the second group into southeast Idaho samples (Group F) and eastern Washington samples (Group G). sNMF analysis indicated the best value for *K* was two (Supplementary Fig. 2c), and separated samples into the same two groups identified on the first PCA axis (Fig. 4c). Pairwise $F_{ST}$ values were statistically significant ($p < 0.01$) between the groups that separated on the first PCA axis ($F_{ST} = 0.304$ and 0.335), but not between the groups that separated on the second PCA axis (Table 2). Notably, sample sizes were low for Group F ($n = 3$) and Group G ($n = 7$).

For *L. canus*, specimens had been collected from three plots that had received different pesticide treatments across a single agricultural field in Hermiston, Oregon (described further below). PCA revealed no evidence for genetic divergence between agricultural plots; the first and second axes explained only 2.33% and 1.84% of the variation, respectively, and no clustering by pesticide treatment type was evident on the PCA plot (Fig. 4c). sNMF analysis indicated the best $K = 1$ (Supplementary Fig. 2d), further supporting the lack of genetic divergence across agricultural plots. In addition, pairwise $F_{ST}$ values between the three plots were non-significant (Table 2). The single Washington *L. canus* sample (which had been morphologically identified as *L. californicus*) did not pass the missing data filters for the within-species analysis.

To test for more fine-scale geographic structure, we also conducted PCA independently for each of the distinct groups identified by PCA for *L. californicus* (Groups A–D) and *L. infuscatus* (Groups E–G). Sample sizes were low for most geographic locations in this analysis, but nonetheless samples from the same geographic location sometimes clustered together for both species (Supplementary Fig. 3 and Supplementary Fig. 4). The most distinctive clusters occurred in southeastern Idaho (Group C), where Ririe and Rexburg each clustered separately from the rest of southeastern Idaho (Supplementary Fig. 3c). This clustering did not correspond with crop type (alfalfa, barley, corn, wheat, or sawflower), irrigation type (irrigated or dry), or pesticide treatment type (Supplementary Data 1). sNMF and pairwise $F_{ST}$ analyses were not conducted for analyses of fine-scale geographic structure due to low sample sizes.

**Genetic diversity**. Observed heterozygosity ($H_o$) ranged from 0.209 to 0.437 across the groups identified with PCA and sNMF, nucleotide diversity ($\pi$) ranged from 0.022 to 0.049, and inbreeding coefficients ($F_{IS}$) ranged from −0.343 to 0.058 (Table 3). Notably, some of these estimates may be impacted by low sample sizes ($n < 15$ for five groups) and the presence of fine-scale genetic structure within groups (see previous section). For most groups, $H_o$ was higher than expected heterozygosity ($H_e$), and $F_{IS}$ was negative. The two exceptions to this pattern were *L. californicus* Groups B and C, for which $H_o$ was less than $H_e$, and $F_{IS}$ was positive.

**Estimates of relatedness**. The identification of close relatives within a dataset can provide insight into fine-scale dispersal patterns, since the identification of close relatives in different geographic locations indicates dispersal by at least one individual. Pairwise comparisons between *L. canus* samples collected from the agricultural field in Hermiston, Oregon indicated that 12 pairs of individuals had high relatedness values for all except one of the seven relatedness estimators; here we report results for the triad likelihood estimator[29]. Relatedness ($r$) ranged from 0.246 to 0.579 for the highly related pairs (Supplementary Fig. 5, Supplementary Table 1). For ten of these highly related pairs, both individuals were sampled in the same plot within 2 weeks of each other. For one pair, the two individuals were sampled in the same plot 75 days apart ($r = 0.406$). For the final pair, the two individuals were sampled in two separate plots (semi-conventional and semi-organic) 302 days apart ($r = 0.289$) (Supplementary Table 1).

**Selection across agronomic treatments**. All specimens collected in Hermiston, Oregon were *L. canus* and were collected from three plots at the OSU Hermiston Agriculture Research and Extension Center in 2014 and 2015 for which clover and potato were grown in alternating years. Each plot had received different pesticide treatments: (1) conventional plot: sprayed with broad spectrum pesticides from 2010 to 2015; (2) semi-conventional plot: treated with broad spectrum pesticides from 2010 to 2011, but pesticide-free from 2012 to 2015; (3) semi-organic plot: treated with broad spectrum pesticides from 2008 to 2009, but only treated with sprays certified by the Organic Materials Review Institute from 2010 to 2015.

$F_{ST}$ outlier analyses for *L. canus* in Hermiston, Oregon were conducted to identify SNPs putatively influenced by different selective pressures in agricultural plots with different pesticide treatments. Stringent correction for multiple tests ($q < 0.05$ for

**Table 3 Sample sizes, observed heterozygosity ($H_o$), expected heterozygosity ($H_e$), inbreeding coefficient ($F_{IS}$), and nucleotide diversity ($\pi$) for genetically divergent groups within species.**

| Species | Population | n | $H_o$ | $H_e$ | $F_{IS}$ | $\pi$ |
|---|---|---|---|---|---|---|
| L. californicus | Group A | 23 | 0.260 | 0.250 | −0.065 | 0.033 +/− 0.016 |
| | Group B | 6 | 0.349 | 0.365 | 0.058 | 0.039 +/− 0.020 |
| | Group C | 33 | 0.209 | 0.220 | 0.028 | 0.034 +/− 0.016 |
| | Group D | 13 | 0.377 | 0.343 | −0.107 | 0.022 +/− 0.011 |
| L. canus | Conventional | 37 | 0.258 | 0.239 | −0.086 | 0.042 +/− 0.020 |
| | Semi-conventional | 78 | 0.220 | 0.205 | −0.093 | 0.041 +/− 0.020 |
| | Semi-organic | 28 | 0.268 | 0.251 | −0.090 | 0.041 +/− 0.020 |
| L. infuscatus | Group E | 11 | 0.402 | 0.340 | −0.343 | 0.035 +/− 0.017 |
| | Group F | 3 | NA | NA | NA | NA |
| | Group G | 7 | 0.437 | 0.367 | −0.287 | 0.049 +/− 0.025 |

$H_o$, $H_e$, and $F_{IS}$ were only calculated for groups with $n > 5$.

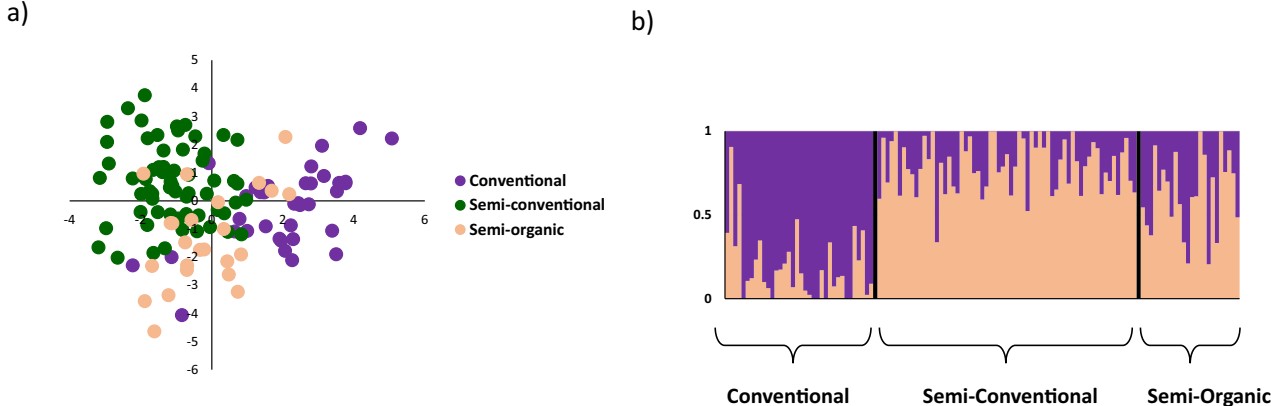

**Fig. 5 Genetic structure for Hermiston, Oregon L. canus using only the 249 SNPs identified as $F_{ST}$ outliers. a** PCA; first and second axes account for 7.32 and 5.54% of variation, and **b** sNMF ancestry proportions for $K = 2$.

Bayescan and OutFLANK, Bonferroni correction for FDIST) resulted in one SNP identified as an outlier with FDIST, and no SNPs identified as outliers for Bayescan or OutFLANK. However, these multiple test corrections are highly stringent for $F_{ST}$ outlier tests, which tend to have low power because signals of selection are often weak, and therefore we also investigated SNPs that were $F_{ST}$ outliers with $p < 0.01$ for OutFLANK and FDIST (Bayescan does not output $p$ values). Under these criteria, a total of 133 outlier SNPs were identified with OutFLANK, and 244 outlier SNPs with FDIST (Supplementary Data 2). Of the 133 SNPs identified as outliers by OutFLANK, all except five were identified as outliers by both tests, resulting in a total of 249 outlier SNPs across both tests. These 249 outlier SNPs were distributed across 163 scaffolds of the reference genome, with no indication of genomic islands of differentiation (Supplementary Data 2, Supplementary Data 3). PCA using only SNPs identified as outliers (Fig. 5a) separated the three treatments to a greater extent than did the PCA with all SNPs (Fig. 3b), with the first and second axes explaining 7.32% and 5.54% of the genetic variation, respectively. sNMF cross-entropy values varied substantially across runs (Supplementary Fig. 6), but the runs with the lowest cross-entropy values primarily separated the conventional treatment from the other two treatments (Fig. 5b). Pairwise $F_{ST}$ values with outlier SNPs were all highly significant ($p < 0.00001$) and were highest for comparisons involving the conventional plot ($F_{ST} = 0.071$ for conventional versus semi-conventional; $F_{ST} = 0.070$ for conventional versus semi-organic; $F_{ST} = 0.049$ for semi-conventional versus semi-organic).

**Phylogenetic analysis**. We obtained genome skimming sequence data from 26 samples, including representative samples from each of the major distinct groups identified using RADseq (L. canus, L. californicus Groups A–D, and L. infuscatus Groups E and F/G). NJ trees created using 488 bp COI sequences and 297 bp 16S sequences from these samples and other Limonius sequences from GenBank identified four L. californicus lineages that corresponded in geographic distribution with RADseq Groups A–D, and two L. infuscatus lineages that corresponded with RADseq Groups E and F/G (Fig. 6, Supplementary Fig. 7). Most of the GenBank samples were from the same geographic locations as our samples, and all samples collected from the same geographic location fell into the same mtDNA lineage (described in detail in Supplemental Results).

The phylogenetic tree based on full-length COI and 16S data showed relatively short branch lengths for Groups A–F/G, indicating that these groups diverged more recently than the currently described Limonius species (Fig. 7). Low maximum likelihood bootstrap values between groups in the L. californicus complex indicated that divergence occurred around the same time period for these four groups. Bayesian evolutionary analysis placed the divergence times for L. californicus Groups A–D between 833,000 and 1.88 million years ago, and the divergence times for L. infuscatus Groups E and F/G between 500,000 and 901,000 years ago (using an estimated COI divergence rate for insects of 3.54% per million years)[30] (Supplementary Fig. 8). COI divergence between L. californicus Groups A–D ranged from 4.2 to 5.3%, and COI divergence between L. infuscatus Groups E and F/G was 2.2%.

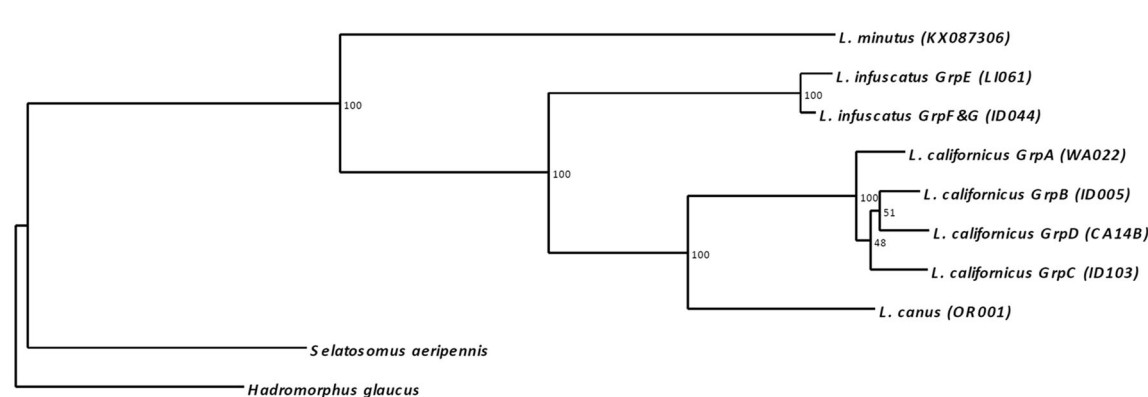

**Fig. 6 Neighbor Joining tree of 488 bp *COI* sequences from this study (*Limonius* samples and two outgroups, *Hadromorphus glaucus* and *Selatosomus aeripennis*) and from *Limonius* samples retrieved from GenBank from other studies.** For sequences from this study, sample names are included, along with the RADseq group if RADseq data were collected for the sample. Sample names for sequences from other studies include Genbank accession numbers; samples for which the species name was not reported on GenBank are indicated by "sp.".

**Fig. 7 Phylogeny of four described *Limonius* species, including potential cryptic species complexes within *L. californicus* and *L. infuscatus*.** The best Maximum Likelihood tree is shown with bootstrap values from 1000 replicates.

## Discussion

Combined RADseq and mtDNA analyses identified strong genetic divergence between the three described *Limonius* wireworm species and evidence that *L. californicus* and *L. infuscatus* may each be comprised of cryptic species complexes. Although no definitive criteria exist for species delimitation based on genetic divergence, within-species *COI* divergence is typically <1% and rarely >2% for animals[31–34]. Both *L. californicus* and *L. infuscatus* contain groups with distinct *COI* lineages exceeding 2% divergence (4.2–5.3% between four groups within *L. californicus*, and 2.2% between two groups within *L. infuscatus*). Furthermore, each of these groups was strongly divergent at RADseq loci ($F_{ST}$ ranging from 0.251 to 0.491). The geographic distributions of these groups also provides support for the possibility of distinct species incapable of interbreeding; although most of the *L. californicus* groups were separated by mountain ranges, two groups in southern Idaho are distributed geographically close together and are separated by no obvious barriers to gene flow, raising the possibility that these two groups come into contact without interbreeding (Fig. 1a). In addition, the two *L. infuscatus* groups have overlapping distributions in Spokane, indicating that these two groups likely come into contact but do not interbreed (Fig. 1b). However, laboratory tests would be required to confirm whether these groups are capable of interbreeding, and detailed morphological assessments would be needed to determine whether morphological differences exist. Furthermore, criteria for species designations are not well-established for click beetles, and the level of *COI* divergence between the groups described here is lower than that for most described *Limonius* species (Fig. 6). Hereafter we refer to these groups as putative cryptic species, while acknowledging that the groups could also be considered strongly divergent populations within species.

The relatively shallow genetic divergence and morphological similarity between the putative cryptic species within the *L. californicus* and *L. infuscatus* complexes indicate that reproductive isolation of these groups occurred more recently than for most of the currently recognized *Limonius* species. Molecular dating placed the divergence times for the cryptic species in the Pleistocene. Many other animal and plant species within this geographic region have phylogeographic structure that dates to this time period, and this structure is thought to be driven by reproductive isolation in Pleistocene glacial refugia, followed by range expansions after glaciers retreated[35–37]. For *L. infuscatus*, post-Pleistocene range expansions are a likely cause of the modern-day overlapping distribution for the two cryptic species. For *L. californicus*, most of the cryptic species are currently separated by mountain ranges, which likely act as strong modern-day barriers to gene flow for both adults and larvae due to a lack of suitable habitat. However, the south-central Idaho and southeast Idaho species are separated by no obvious geographic barriers to gene flow, aside from the Craters of the Moon lava field which formed 2000–15,000 years ago.

The presence of two exceptions to the major *L. californicus* groupings potentially indicate rare hybridization events between cryptic species. In PCA plots, one individual from Columbia, Washington plotted approximately midway between the clusters of Groups A and B (Fig. 4a); this result is surprising because Columbia is well within the Group A geographic distribution and distant from the Group B geographic distribution, although further sampling could indicate that Group B has a wider distribution than documented here. This individual could potentially be a hybrid resulting from a long-distance dispersal event, possibly mediated by humans as a result of agricultural activity. In addition, one individual from Townsend, Montana plotted outside the main cluster for Group C (Fig. 4a). This sample may simply represent a separate population within

Group C; the geographic location of this sample (Townsend, Montana) may harbor a distinct population for Group C due to its relatively long geographic distance from the rest of our Group C samples, as well as its separation by mountains. Larger sample sizes from this location would be needed to evaluate this hypothesis.

The genetic structure identified here for *L. californicus* and *L. infuscatus* aligns with previous studies. For *L. californicus*, the four distinct groups clustered both geographically and genetically with samples analyzed in two previous *COI* and *16S* DNA barcoding studies[17,18]. These previous studies did not include samples from the geographic region where the fourth distinct group in our study occurred (south-central Idaho). For *L. infuscatus*, the two mtDNA lineages were also present in a previous study that included samples from a similar geographic region, although most of the samples from that study were collected east of ours, extending into western Washington[18].

Knowledge of the existence of cryptic species is important to IPM because different species respond differently to pest management strategies (reviewed in ref. [10]). However, the presence of morphologically similar cryptic species exacerbates the challenges of distinguishing wireworm species. For the *L. californicus* species complex, the geographic origin of the samples can be used to help distinguish species, because each cryptic species occurs in a different geographic region. In contrast, the cryptic *L. infuscatus* species have overlapping geographic distributions, and therefore geography will be less useful for distinguishing species. We did not conduct morphological analyses to look for diagnostic differences between the cryptic species identified here, but Etzler et al.[18] stated they found no morphological differences between the three putative *L. californicus* species found in their study. Efforts have yet to be initiated to morphologically characterize these putative species. Until then, DNA barcoding remains the most efficient and effective method for distinguishing these species.

We found evidence for genetically divergent populations separated by relatively short geographic distances for many of the cryptic species, e.g., as little as 25 km for one of the *L. californicus* species (i.e., Ririe, Idaho vs. Rexburg, ID for Group C) and 80 km for one of the *L. infuscatus* species (i.e., Lincoln, WA vs. Spokane, WA for Group E) (Supplementary Figs. 3 and 4). Genetically distinct groups did not correspond with crop type, irrigation type, or pesticide treatment type for agricultural fields where individuals were sampled. These results suggest that long-distance movement may be rare for these species, and that variation in many agricultural practices may not impact dispersal patterns. However, per-site sample sizes were relatively low for these analyses, and therefore future studies should explore fine-scale population structure with larger sample sizes.

For *L. canus*, we investigated fine-scale genetic structure across a single agricultural field in Hermiston, Oregon. Population structure analyses found no evidence for genetically distinct groups within the field, indicating the presence of dispersal across the field. However, relatedness analysis indicated that most pairs of highly related individuals were from the same plot within the field, providing evidence for some restrictions in gene flow across the field, although this may be true only for the larval stage. We found only one related pair in different plots, and the two individuals from this pair were sampled almost a year apart. The $r$ value for this pair was 0.289, which indicates a relationship such as half-sibling, grandparent/grandchild, or aunt/uncle-nephew/niece. The presence of a pair of relatives in different plots indicates at least one dispersal event, but does not provide information about whether the dispersing individual was a larva or an adult; based on the much higher dispersal potential for adults than larvae, this individual was likely an adult.

Our results are consistent with previous studies investigating dispersal patterns of larval and adult forms of click beetles (reviewed in ref. [1]). These studies indicate that larval movement is very limited; mark-recapture and stable isotopes analyses indicate that lifetime horizontal movement for individual larvae is likely limited to about 1.5 m[38,39]. Adults have much higher dispersal; for example, mark-recapture and stable isotopes analyses have found mean lifetime adult male dispersal of about 194 m for *Melanotus sakishimensis* in Japan[40], >80 m maximum dispersal for adult male *Agriotes obscurus* in Western Austria[41], and >30 m dispersal in <24 h for adult *A. obscurus* in British Columbia[42]. These estimated adult dispersal distances are low enough to be consistent with our evidence for genetically distinct populations separated by >25 km. However, most prior studies of adult dispersal have not been able to quantify the mean maximum extent of movement, and many have no information regarding female dispersal. Furthermore, mark-recapture studies have found that dispersal rates differ across sexes, habitats, and species[42,43], indicating that results from one study system may not be highly predictive of another study system. Population genetic approaches have the potential to efficiently and effectively increase our knowledge about population structure and dispersal patterns for each agronomically important click beetle species.

Genome scans for wireworm in three plots differing in agronomic treatments within the same agricultural field revealed evidence for adaptive genomic differences driven by treatment type. The three treatment types differed in the length of time since treatment with broad spectrum pesticides: all plots had been treated with broad spectrum pesticides for at least 2 years, but two of the plots had subsequently been free of broad spectrum pesticides for 2 years (semi-conventional plot) or 4 years (semi-organic plot) prior to sampling. These pesticides were aimed at controlling a wide variety of pest species, but not specifically aimed at wireworms, which would have required seed treatments prior to planting. However, prior studies have demonstrated that these pesticide sprays can influence the species composition of adult click beetle populations, although the underlying biological mechanism is unknown[44]. Our genome scans indicated the conventional plot, which had received continuous pesticide treatment for 2 years prior to and during sampling, was the most divergent from the rest of the plots for $F_{ST}$ outlier loci, suggesting that genomic composition changed in response to the discontinuation of pesticide treatment for the non-conventional plots. This genomic change could potentially have been driven by a release from adaptive pressures that were present during pesticide treatment. Although estimating the exact age of the *L. canus* was not possible, all of the collected samples were determined to be later larval instars based on body length. Since the larval stage of *L. canus* lasts for multiple years[45], the wireworms were likely exposed to insecticides for extended periods. Therefore, it is conceivable that the mechanism driving the genetic signature observed here is high mortality in earlier life stages of larvae maladapted to broad spectrum pesticides. This result provides evidence that wireworms can evolve in response to pesticide treatment, which could potentially lead to a reduction in pesticide effectiveness over time. However, the outlier loci use for these analyses were only statistically significant outliers for a relatively lenient significance threshold ($p < 0.01$), whereas stringent correction for multiple testing resulted in only one locus identified as an outlier. We considered loci with $p < 0.01$ to be potentially biologically meaningful since $F_{ST}$ outlier tests frequently have low power because selection often acts on multiple genes, with each gene contributing a small effect to the phenotype and therefore experiencing relatively weak selection. In addition, the short length of time since the change in pesticide treatment in our study system may have been insufficient for a strong

signature of selection to develop, even if selection has been ongoing. Therefore, future studies should further explore the association of these outlier loci with agronomic treatment. In particular, as more genomic resources become available for click beetles, including annotation of the reference genome used here, the genes associated with outlier loci in our study should be identified, and the response of these genes to pesticide treatment should be tested in additional controlled experiments.

In conclusion, genetic analyses of three wireworm species identified putative cryptic species, population structure, and putative genomic adaptation to pesticide treatment. This information provides a critical starting point for evaluating whether IPM strategies differ in effectiveness across populations and species and over time. To our knowledge this is the first study to use a genomic approach to investigate phylogenetic and population genetic structure for wireworms, and our results illustrate the power of these techniques. Future genetic studies are likely to find additional cryptic wireworm species; for example, previous wireworm DNA barcoding studies found evidence for cryptic species complexes within *Hypnoidus bicolor* Eschscholtz and *Hadromorphus glaucus* Germar[17,18]. Future studies investigating population structure should use increased within-site sample sizes and geographic extent of sampling, and could compare detailed habitat information with population structure using "landscape genomics" approaches to investigate the ecological factors driving dispersal patterns or adaptation[46]. To further understand the potential for adaptation driven by pesticides, more controlled pesticide treatment experiments should be conducted across populations and species. In addition, studies investigating adaptation would benefit from increased genomic resources, including annotated whole genome assemblies for each of the agriculturally important wireworm species. Genetic and genomic tools have strong potential but are currently nearly untouched as a resource for wireworm management.

## Methods

**Sample collection.** Wireworms were collected using bait traps between 2012 and 2016 from wheat, potato, clover, barley, alfalfa, corn, and safflower agricultural fields in the northwest US (Washington, Oregon, Idaho, Montana) and southwest Canada (Alberta and Saskatchewan) (Fig. 1, Table 1, Supplementary Data 1). Bait traps were created by burying a water-soaked seed mixture (wheat, barley, and/or corn seeds) ~15 cm underground for about 2 weeks. Samplings were conducted during the growing season from April through September of each year. Prior to generating genomic data, an initial species designation was assigned to each specimen based on visual morphological assessment according to Rashed et al.[22] and Milosavljevic et al.[47] using a dissecting microscope. For subsequent analyses, we selected specimens identified as *L. californicus*, *L. infuscatus*, *L. canus*, or an unknown *Limonius* species. We also selected one specimen identified as *Hadromorphus glaucus* Germar, and one specimen identified as *Selatosomus aeripennis* Kirby, to use as outgroups for phylogenetic analysis. As described above, all specimens collected in Hermiston, Oregon were *L. canus* and were collected at the OSU Hermiston Agriculture Research and Extension Center in 2014 and 2015 from three plots receiving different pesticide treatments. Pesticide sprays were aimed at controlling a variety of pests, but not specifically aimed at wireworms, which would typically require seed treatment prior to planting. The three plots were randomized completed plots 9.14-m long and 7.62-m wide and separated by a 3 m buffer.

**Laboratory analyses.** To obtain sequencing data for de novo assembly of the *L. californicus* genome, genomic DNA was extracted from a specimen collected in Aberdeen, ID as previously described[48]. Genomic DNA libraries were then prepared for Illumina shotgun sequencing, Illumina synthetic long-read (SLR) sequencing[49], and Pacific Biosciences (PacBio) long-read sequencing. For Illumina shotgun sequencing, DNA was sheared to 800 bp using a Covaris M220 focused-ultrasonicator, and then Illumina sequencing adapters were ligated using a KAPA Biosystems ligation kit. The library was sequenced with an Illumina MiSeq v3 600 cycle sequencing kit at the University of Idaho IBEST Genomics Resources Core. An aliquot of the library was also shipped to the University of California Berkeley QB3 Vincent J. Coates Genomics Sequencing Laboratory (QB3 Genomics) for sequencing on a single lane of Illumina HiSeq 2500 with 2 × 250 bp reads. Illumina SLR libraries were prepared using the Illumina protocol for the same DNA sample to help resolve complex repetitive elements within the genome[50]. Briefly, DNA was

sheared to ~10 Kbp fragments, ends were blunted, adapters ligated, DNA was diluted into a 384-well plate, and then long-range PCR was used to amplify the fragments within each well. Each well was then prepared for sequencing using Nextera-based fragmentation with unique barcodes and sequencing adapters added via PCR, and libraries were sequenced on a single lane of Illumina HiSeq4000 with 2 × 100 bp reads at QB3 Genomics. For PacBio long-read sequencing, genomic DNA from the same sample was cleaned and concentrated using Beckman Coulter AMPure beads. DNA was then shipped to the University of Washington PacBio Sequencing Services core, where libraries were prepared and sequenced on four PacBio RSII SMRTcells.

To generate RADseq and genome skimming data, genomic DNA was extracted from the anterior end of each wireworm using Qiagen DNeasy Blood and Tissue Kits, Zymo Quick-DNA Universal Kits, or cetyltrimethylammonium bromide (CTAB) according to Marzachi et al.[51]. RADseq libraries were prepared following the protocol described in ref. [52]. Briefly, genomic DNA was digested using the restriction enzyme *SbfI*, and biotinylated adapters were ligated that contained a restriction cut site overhang and an 8 bp barcode sequence unique to each individual sample. Barcoded samples were then combined into a total of five multiplexed pools, with approximately 96 samples per pool. Pools were mechanically sheared to ~400 bp, and DNA fragments containing biotinylated adapters were captured using Streptavidin beads. The NEBNext Ultra DNA Library Prep Kit for Illumina protocol was then used for each pool, excluding the initial fragmentation step. Sequencing of RADseq libraries was conducted on one lane of an Illumina HiSeq4000 at QB3 Genomics with 150 bp paired-end reads.

To obtain mitochondrial DNA sequences for phylogenetic analysis, shotgun sequencing libraries were created for genome skimming analysis from a total of 26 samples, including samples collected across the geographic sampling range for each *Limonius* species, and for samples from two additional wireworm species from a different genus to use as outgroups (*S. aeripennis* and *H. glaucus*) (Supplementary Data 1). Shotgun libraries were created using a reduced volume Nextera XT protocol[53] and were sequenced on an Illumina MiSeq at the University of Idaho Genomics Resources Core using 600 cycle Reagent Kit v3 (MS-102-3003) with 300 bp paired-end reads.

**Genome assembly.** Synthetic long-read assembly was done using the TruSeq Long-Read Assembly application available through BaseSpace. The hybrid assembler MaSuRCA v3.2.4[54] was then used to assemble Illumina, SLR, and PacBio reads into contigs. Illumina MiSeq and HiSeq reads were provided to MaSuRCA without any preprocessing as suggested by the MaSuRCA documentation, and because MaSuRCA allows only one long-read input type, PacBio reads were combined with SLR reads and provided to MaSuRCA as "PACBIO" type data. Aside from NUM_THREADS = 50 and JF_SIZE = 72000000, all other parameters were left at default settings. Assembly was done at the University of Idaho Computational Resources Core. To assess genome assembly completeness, BUSCO v.3[55] was used to search for near-universal genes using the insecta_odb9 single-copy ortholog reference set, which contains 1658 reference genes.

**RADseq filtering and genotyping.** hts_AdapterTrimmer in HTStream (https://ibest.github.io/HTStream) was used to trim Illumina adapters from raw RADseq reads and to remove any remaining reads <50 bp long. Trimmed reads were then cleaned and demultiplexed (i.e., separated by individual barcode) with PROCESS_RADTAGS in STACKS v2.3e[56] using the parameters -c, -q, -r, and --bestrad. PCR duplicates were removed using hts_SuperDeduper in HTStream, and remaining reads were aligned to the *L. californicus* reference genome described above using BWA v0.7.17[57]. We then removed any samples with low numbers of total mapped reads (<50,000) from subsequent analyses.

Joint genotyping was conducted using HaplotypeCaller in GATK v4.1.1.0[58] using default parameters. VCFtools v0.1.15[59] was used to filter indels, SNPs with depth <5 and/or genotyping quality <15, SNPs missing in more than 20% of individuals, individuals with more than 20% missing data, SNPs with low mean depth (<8) or high mean depth (>1.5 standard deviations above the mean depth of all SNPs), and SNPs that were singletons or monomorphic. We performed these filtering steps for the full dataset of *Limonius* samples, and also independently for each known species, as well as for each subgroup within known species. Performing these filtering steps independently for each species and subgroup allowed us to maximize the performance of the within-species and within-subgroup analyses by maximizing the number of SNPs retained, minimizing the amount of missing data, and ensuring singletons and monomorphic SNPs were removed. Notably, performing these steps independently for each data subset resulted in different numbers of SNPs across analyses, as well as slight differences in sample sizes across analyses due to different numbers of individuals passing the missing data filters. For *L. californicus*, high numbers of sequence reads per sample allowed us to use a more stringent filtering criterium of removing SNPs with any missing data, in addition to the other filters described above.

**Population structure and diversity.** We evaluated population structure using PCA and sNMF ancestry coefficient estimation using LEA v.2.4.0[60] in R v.3.5.1[61]. For the PCA including all three described *Limonius* species, we had strong differences in sample sizes across species; therefore, we conducted the PCA with the

full dataset, and also with a dataset randomly subsampled to equal sample sizes across species, since uneven sample sizes can strongly bias PCA[28]. For sNMF analysis, we ran five iterations for each value of *K* ranging from 1 to 10, and chose the best *K* by evaluating cross-entropy values across *K*. We conducted these analyses using the full dataset with all three species, and also independently for each species to investigate within-species population structure. We then evaluated the level of genetic divergence between each distinct group identified by PCA and sNMF by calculating pairwise $F_{ST}$ values using Arlequin v. 3.5.2.2[62], testing for significance with 10,000 permutations, and by calculating Nei's genetic distance using Adegenet v.2.1.1[63]. We also calculated observed heterozygosity ($H_o$), expected heterozygosity ($H_e$), nucleotide diversity ($\pi$), and the inbreeding coefficient ($F_{IS}$) for each genetically distinct group using Arlequin.

**Relatedness analysis using *L. canus*.** To investigate whether close relatives were present in the dataset, we estimated relatedness (*r*) between all pairs of individuals using seven relatedness estimators using Coancestry v.1.0.1.8[64]. Values of *r* decrease with decreasing relatedness; for example, full siblings or parent-offspring should have *r* ~ 0.5, and half siblings or aunt/uncle-niece/nephew should have *r* ~ 0.25. We conducted these analyses for the *L. canus* samples collected from Hermiston, Oregon; the other genetically distinct groups in this study had insufficient sample sizes for this analysis. We used a subset of SNPs that had been "thinned" to minimize linkage for this analysis by removing SNPs closer than 1000 bp apart in the genome.

**Selection across agronomic treatments.** To identify SNPs putatively under divergent selective pressures across agronomic treatment types, we conducted $F_{ST}$ outlier tests for *L. canus* samples collected from the three plot types within the agricultural field in Hermiston, Oregon. To identify $F_{ST}$ outliers, we used three approaches: OutFLANK[65], Bayescan[66], and FDIST[67] as implemented in Arlequin. OutFLANK uses a likelihood approach to infer the distribution of $F_{ST}$ for neutral markers, Bayescan uses a Bayesian approach based on the multinomial-Dirichlet model, and FDIST uses a coalescent approach to infer the distribution of $F_{ST}$ as a function of heterozygosity for neutral markers. Corrections for multiple testing were conducted by controlling the false discovery rate (FDR) for Bayescan and OutFLANK, and using Bonferroni correction for FDIST. To investigate the level of genetic divergence between agricultural plots for the outlier SNPs, we conducted PCA, sNMF, and pairwise $F_{ST}$ analysis using only outlier SNPs.

**Mitochondrial DNA assembly and phylogenetic analysis.** Full *COI* and *16S* rDNA sequences were obtained using a genome skimming approach[26,27]. Sequence reads were first cleaned to remove PCR duplicates, Illumina PhiX control reads, sequencing adapters, uncalled bases, and low quality bases using HTStream (https://github.com/ibest/HTStream). An iterative mapping and de novo assembly strategy, ARC (https://github.com/ibest/ARC), was then used with *COI* and *16S* sequences obtained from a published *L. californicus* mitogenome (Genbank KT852377.1[48]) as mapping references. ARC was run with the following parameters: numcycles=2, mapper=bowtie2, assembler=newbler, sloppymapping=False. The resulting contigs were filtered and oriented using BLAT (Kent 2002) and custom Python scripts (see https://github.com/kimandrews/Wireworm_popgen).

To investigate phylogenetic relationships between the samples included in this study and other *Limonius* species, we downloaded all available *Limonius COI* sequences from GenBank and aligned these with our full *COI* sequences using MUSCLE[68] in Geneious 9.1.8 (https://www.geneious.com). We then trimmed all sequences to the same length, resulting in a 488 bp fragment for a total of 380 GenBank samples from 19 *Limonius* species. We conducted the same procedure for the *16S* sequence data, resulting in a 297 bp fragment for a total of 52 samples from six *Limonius* species. In addition, we included *COI* and *16S* sequences from whole mitogenomes for *L. californicus* (GenBank No. KT852377.1[48]) and *L. minutus* (GenBank No. KX087306.1). We created neighbor joining (NJ) trees for each marker using GENEIOUS, with *H. glaucus* and *S. aeripennis* samples used as outgroups. Using these results, we identified putative cryptic species that had distinct *COI* and *16S* lineages and were also strongly divergent for RADseq data. We then conducted maximum likelihood phylogenetic analyses using RAxML[69] with concatenated full-length *COI* and *16S* sequence data for one individual from each known species and each putative cryptic species, with sequence data partitioned by locus. We first identified the best-scoring tree from 50 maximum likelihood searches on distinct starting trees, and then performed 1000 bootstraps on the best tree. We then used BEAST2 v.2.6.2[70] to estimate divergence times using the full-length *COI* sequences, using an estimated *COI* divergence rate available for insects (3.54% divergence between lineages per million years[30]). We used the GTR nucleotide substitution model with log normal GTR rate priors, based on AIC results from jModelTest v2.1.10[71], which identified the best model as GTR + I + G. We used the Yule Model and a strict molecular clock, conducting a run of 10 million generations, sampling every 1000 generations after a 10% burn-in. We used Tracer v.1.7.1[72] to assess convergence based on ESS values and visual inspection of trendlines, and used TreeAnnotator v.2.6.2 to generate the maximum clade credibility tree.

**Statistics and reproducibility**. This study used 247 wireworm specimens from 29 geographic locations across the northwest US and southeast Canada. Descriptions of all statistical analyses and *p* values are provided in the text. Commands and scripts used are provided at https://github.com/kimandrews/Wireworm_popgen.

**Reporting summary**. Further information on research design is available in the Nature Research Reporting Summary linked to this article.

## Data availability

The sequence data from this study are available on NCBI (https://www.ncbi.nlm.nih.gov). The Whole Genome Shotgun project has been deposited at DDBJ/ENA/GenBank under the accession JABXWE000000000. The version described in this paper is version JABXWE010000000. The genome assembly, raw sequence reads for the genome assembly, and RADseq reads are under Project number PRJNA595620. The GenBank accession numbers for the *COI* sequences are MT571672-MT571696 and for the *16S* rDNA sequences are MT578070-MT578094.

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

## Acknowledgements

We would like to thank members of Idaho IPM Laboratory for collecting samples. Data collection and analyses performed by the IBEST Genomics Resources Core at the University of Idaho were supported in part by NIH COBRE grant P30GM103324. This work used the Vincent J. Coates Genomics Sequencing Laboratory at UC Berkeley, supported by NIH S10 OD018174 Instrumentation Grant. Funding was provided by the Idaho Wheat Commission project number 6625 to S.S.H., A.G., and A.R.; Idaho Barley Commission project number AN2685 to A.R.; USDA-NIFA-Hatch-IDA01506 to A.R.; and Oregon Potato Commission project number ARF 7252 for the field trial to S.I.R. This publication was made possible by the NSF Idaho EPSCoR Program and by the National Science Foundation under award number OIA-1757324. Publication of this article was partially funded by the University of Idaho - Open Access Publishing Fund.

## Author contributions

S.S.H., A.G., K.R.A., A.R., S.I.R., C.M.W., and P.A.H. designed the study. K.R.A. and S.S.H. analyzed the data. A.R., D.W.C., S.I.R., W.G.V.H., K.W.W., and R.V. provided wireworm samples. A.G., D.D.N., and M.W.F. performed laboratory work. K.R.A. and S.S.H. wrote the manuscript. All authors reviewed the results and approved the paper.

## Competing interests

The authors declare no competing interests.
