## [Peer Review File · Communications Biology]

Reviewers' comments:

Reviewer #1 (Remarks to the Author):

Here, Andrews et al. present a reference genome, RADseq-based population genomic analysis, and genome-skimming-based phylogeny of several pest wireworm species in western North America. From these analyses, they discover patterns of cryptic diversity, within-species population structure, and potential signs of genomic adaptation associated with pesticide use. Overall, the paper is very well-written, and I found virtually no grammar/orthography issues. The methods and analyses are all on-point and the conclusions are appropriate and justified. I really only have some minor revisions to suggest, and one broader comment about the organization of results before methods that could entail some additional editing. This paper appears to be one of the first population genomic studies of pest click beetles, and provides novel insight into genetic differentiation and divergence of these insects.

The authors do a great job of superficially introducing the approach of the current study at the end of the introduction, which is helpful given the presentation of results before methods in this journal. One thing I find difficult with this organization of papers (i.e. this is not a criticism of this manuscript at all, but more the results before methods organization), is how much methodological detail needs to be included in the results so that the paper stands on its own when read front to back. Perhaps this is overthinking it, and it should be assumed that readers are constantly flipping back forth between methods and results, but several of my comments refer to this kind of detail in the results. Feel free to ignore those comments if they are overreacting to results before methods organization, however it seems that this manuscript was likely written with methods before results, and then edited to conform to this journal's organization. Again, I don't mean this negatively, as I have done the same many times, but I wonder if just a bit more shuffling of details from the methods to the results would help in presentation for readers who would read this manuscript from start to finish.

SPECIFIC COMMENTS

Line 70. Should "i.e." be replaced with "e.g." here?

Line 86. "Enhanced" sounds overly positive for this sentence—I would suggest "furthered" or "exacerbated".

Line 118. Please provide authorities for these species.

Line 146. Is there a flow cytometry estimate for genome size of this species, or closely related species?

Lines 153-154. What are the distributions of these species in North America? This would be potentially important in interpreting the divergence dating analyses with regard to Pleistocene glaciations as well as short to long-term geographic isolation between species/lineages. Perhaps this could be incorporated into Figure 1?

Line 162. I am unsure what is being mentioned with the "as described above" here. Is this a hold-over from a draft with methods before results?

Line 171. Given the presentation of results before methods, I would like to see a brief description here about how species identification was conducted, or at least cite the same refs as in the methods.

Line 220. While I understand the reference to "within species" analyses/datasets, relative to lines 160-161, it is unclear whether these were used for the within species analyses (3 paragraphs after line

183) or whether the full dataset was used. Perhaps it would be useful to either clarify within these paragraphs or add another sentence around line 160 to clarify what data is being used in what analyses.

Lines 318-327. I am not too big a fan of using a straight divergence rate to estimate ages, even if the age estimates are not a large part of this study. I'd be curious what the estimated divergence ages would be using a molecular dating approach, even just with secondary calibrations based on recent phylogenomics studies (e.g. 10.1073/pnas.1909655116) or some of the other older studies based on a few more comparable genes. I'm not saying that this is 100% necessary for this study, it might be worth exploring if other reviewers make similar comments.

Lines 464-466. This is an important point, and I think could be expanded on. Was/is it possible to estimate the larval instar of these samples, and from that infer age (even as broadly as > or < 5 years old)? Given the long development time, it's possible that all sampled individuals were greater in age than any of the pesticide treatments (max 6 years, if I am interpreting everything correctly)! Alternatively, is the history of these plots known from before the current pesticide application period? I wonder also about the genomic locations of these ~250 SNPs. Of course, with 115k contigs, there's going to be a limit to what you can say about genomic location. But if most of these occur on a handful of contigs, that might be something to report. Now that I see the final supplementary table, it doesn't look like any meaningful pattern there, but maybe that's a point to bring up as well (i.e. "no indication of a genomic island of differentiation" or something).

Line 577. What kind of parameters were used for GATK? Were the "best practices" followed for species without existing genotype data?

Lines 629-630. Was the full HTStream pipeline used here, or just particular parts of it? Are any parameters/settings needed?

Line 635. Make reference to the github page containing Python scripts?

Table 1. Were samples at a particular locality all collected at the same time? Or were localities repeatedly sampled over the 4 year period? I suppose this point only really affects the interpretation of the *L. canus* samples (see comment line 464-466), so maybe it would be sufficient to just specify what year those were collected, either here or in the text (around line 506).

Reviewer #2 (Remarks to the Author):

Andrews et al. present some nice work identifying cryptic species of wireworm and attempting to find outlier SNPs associated with pesticide treatments. I agree with the authors that genomics has been underutilized in many species of agricultural concern and this paper does a nice job of jumping into a system that had no previous genomic resources. I found the paper to be very well written. Further, the authors provide clear descriptions of what they did in terms of data generation, analyses, and their interpretation of results. All data and code appears to be available. I have a few suggestions that I think would improve the manuscript.

Major (in no particular order):

1) I found the mapping results, PCA results, and mtDNA tree to be somewhat at odds. Lines 156-158 say that the lowest average mapping rate was for the *L. infuscatus* samples. This result suggests that

this species is very sequence divergent from the reference and the authors point this out (lines 162-165, and more on that below). However, PC1 of Figure 2 (50% of variation) splits californicus/infuscatus and canus, not californicus/canus and infuscatus as I would expect. Is the low mapping rate not due to sequence divergence? Further, the mtDNA tree (Figure 7) suggests that canus and californicus are more closely related, consistent with the mapping results, not the Figure 2 PCA.

2) Continuation from above. Given F_{st} is so high (nearly 1) between *L. californicus*, *L. canus* and *L. infuscatus*, some metric characterizing percent sequence divergence (e.g., D_{xy}) between species needs to be included in the paper. It also adds to the comparison of sequence divergence calculated from the mtDNA. Nucleotide diversity would be nice as well.

4) Motivation for the relatedness analysis was unclear in the introduction or results, it only became clear when reading the methods at the end of the paper.

5) Why is *L. infuscatus* considered three groups in figure 4b? It seems like it is just two, E and FG. F and G only seem to be splitting apart a slight bit in the PCA due to some sampling/geography pattern that doesn't show up in sNMF analyses (Figure 3) or in mtDNA analyses.

6) Did the authors explore how much of an impact including *infuscatus* had on F_{st} estimates between *canus* and *californicus*. Given the low percentage mapping, it must have heavily biased the SNPs. Asked another way, did the authors compare F_{st} when filtering only *canus* and *californicus*, to the F_{st} value they report that is estimated when filtering all three together.

7) I would suggest the authors consider adding a few sentences about the phylogeographic patterns they see and how it fits with other systems. It seems to me that there are several vertebrate and invertebrate examples that show distinct phylogeographic breaks consistent with the two cryptic species of *L. infuscatus* and the *L. californicus* group A split with the other three groups.

8) Were sex linked SNPs or scaffolds used in the analysis? Do you know the sex of any of the samples? If so, do they correspond to any of the outliers in PCAs or to the F and G groups?

9) I was wondering if the authors could comment on if their outlier SNPs are near, or in, genes with functional relevance with respect to pesticides? Is their reference genome annotated?

10) There seems to be some inconsistency between plots and tables that needs explaining/fixing. For *infuscatus*, Table 2 says there are 3 from group F and 7 from group G (10 total). It also says this in the text (Line 211). However, Figure 4 b shows 11 dots for F + G. Further, figure 3 seems to show 12 bars for F + G. I didn't double check all the others.

Minor:

Figure 7. *L. canus* should be labeled. I would also suggest adding the species name by the groups in the main part of the figure.

Lines 144-145. A few commas are missing for numbers.

Size of the scaffolds for the F_{st} outlier snps would be useful (table S2). Some of these could be in close proximity to one another, especially if they are all located on small scaffolds.

Conclusions statement might need to be more nuanced. I don't believe you conclusively 'identified genomic adaptation to pesticide treatment'.

Figure 4. Consider including species name in corner of each a) b) c) plot. Makes it easier for the reader to remember what species they are looking at without jumping back to text.

Reviewer #3 (Remarks to the Author):

Overall this is a fantastic study that should be accepted for publication. The in-depth study of the different *Limoni* populations and species is a great resource for future IPM efforts in the region, as well as future population studies for these species. These data are presented clearly, and the figures are very informative. The methodology is also presented clearly, allowing for reproduction of these finds, as well as applications to other wireworm species.

One minor typographical issue is a missing period on page 15, line 336. I did not notice any others, but a good recommendation is another skim through by the authors before final publication.

The only issue that I would like to bring up here is more philosophical, rather than a rejection of the conclusions drawn in this work. I would like all of the author's to really consider this alternative interpretation. The DNA data presented is all mitochondrial DNA, yet frequently the authors mention that different populations do not interbreed (pg 16), and also mention one individual as a possible hybrid (pg 17). However, the null hypothesis of mitochondrial DNA inheritance is uniparental inheritance from the mother only. Exceptions to this rule exist, but have not been demonstrated within elaterids (i.e. wireworms). Looking at mitochondrial DNA would not show interbreeding between populations, as only the mother would pass these genes down. There would also be no hybridization or recombination events, as the mitochondrial genome is a single chromosome inherited from a single parent. I argue that these data actually demonstrate the poor dispersal by female *Limoni*. While these data suggest deep divergences between populations, potentially leading to novel traits in each population, we can make no claim about the lack of interbreeding here, as these events would not be seen if only males travel long distances as their mito-genomes would not appear. As most of the references included cite male dispersal, the authors need to consider this possibility. The presence of two nearby female lines of *Limoni infuscatus* within a field argue, in my opinion, that interbreeding is possible

Now, in my personal observations, there are some morphological variations between these populations, suggesting that you are all correct that there may be cryptic species present within *Limoni californicus* and *L. infuscatus*. The half-million to million year divergence time in the mito-genome groups is also indicative of long periods of isolation. However, we can't say for certain that these may be cryptic species yet. It may be that while females disperse poorly, allowing for isolated mito-genome lines in the valleys, but males could be stronger dispersers. The males may move nuclear genes between populations, keeping them all under the umbrella of a single species, despite distinct and isolated female mitochondrial populations. These also may be relicts of glaciation trends at the end of the last ice-age on the populations.

In short, I would like the authors to acknowledge the limitations of interpreting mitochondrial DNA on the possibility of interbreeding between populations. No claims of this can be made due to the nature of mitochondrial inheritance. This actually stresses the need to examine nuclear genes to evaluate the possibility of cryptic species or if these populations interbreed. I look forward to reading that paper when you all get around to that!

The above points have no effect on the validity of publication. The authors are free to reject the above points, but I would like them all to think on them. I think their arguments would be strengthened if they mention them in their discussion of their data .

RESPONSE TO REVIEWERS

Reviewers' comments:

Reviewer #1 (Remarks to the Author):

Here, Andrews et al. present a reference genome, RADseq-based population genomic analysis, and genome-skimming-based phylogeny of several pest wireworm species in western North America. From these analyses, they discover patterns of cryptic diversity, within-species population structure, and potential signs of genomic adaptation associated with pesticide use. Overall, the paper is very well-written, and I found virtually no grammar/orthography issues. The methods and analyses are all on-point and the conclusions are appropriate and justified. I really only have some minor revisions to suggest, and one broader comment about the organization of results before methods that could entail some additional editing. This paper appears to be one of the first population genomic studies of pest click beetles, and provides novel insight into genetic differentiation and divergence of these insects.

The authors do a great job of superficially introducing the approach of the current study at the end of the introduction, which is helpful given the presentation of results before methods in this journal. One thing I find difficult with this organization of papers (i.e. this is not a criticism of this manuscript at all, but more the results before methods organization), is how much methodological detail needs to be included in the results so that the paper stands on its own when read front to back. Perhaps this is overthinking it, and it should be assumed that readers are constantly flipping back forth between methods and results, but several of my comments refer to this kind of detail in the results. Feel free to ignore those comments if they are overreacting to results before methods organization, however it seems that this manuscript was likely written with methods before results, and then edited to conform to this journal's organization. Again, I don't mean this negatively, as I have done the same many times, but I wonder if just a bit more shuffling of details from the methods to the results would help in presentation for readers who would read this manuscript from start to finish.

RESPONSE: We appreciate the reviewer's positive overview of our work, as well as the identification of areas where the Results section could be clarified to make it easier for the reader to follow before reading the Methods section. We believe that addressing the reviewer's specific comments (listed below) have allowed us to achieve that improved clarity. We have addressed all specific comments below.

SPECIFIC COMMENTS

1.1) Line 70. Should "i.e." be replaced with "e.g." here?

RESPONSE: Done. We changed the wording to clarify that the new generation insecticides are neonicotinoids (we prefer not to use "e.g." because neonicotinoids are the new class of insecticides, rather than just an example of the new insecticides). It now reads: "the new generation insecticides are neonicotinoids" (now line 86).

1.2) Line 86. “Enhanced” sounds overly positive for this sentence—I would suggest “furthered” or “exacerbated”.

RESPONSE: Done. We changed this to “exacerbated”

1.3) Line 118. Please provide authorities for these species.

RESPONSE: Done.

1.4) Line 146. Is there a flow cytometry estimate for genome size of this species, or closely related species?

RESPONSE: We did a search but have not been able to find flow cytometry estimates of genome size for any Limonius species.

1.5) Lines 153-154. What are the distributions of these species in North America? This would be potentially important in interpreting the divergence dating analyses with regard to Pleistocene glaciations as well as short to long-term geographic isolation between species/lineages. Perhaps this could be incorporated into Figure 1?

RESPONSE: We now clarify that these species are endemic to the northwest US and southwest Canada (i.e., they are native to this region and only found there), with the exception of L. californicus, which has a distribution additionally extending throughout California (see lines 147-148).

1.6) Line 162. I am unsure what is being mentioned with the “as described above” here. Is this a hold-over from a draft with methods before results?

RESPONSE: Thank you for catching this. We now replaced “as described above” with “see Methods section”

1.7) Line 171. Given the presentation of results before methods, I would like to see a brief description here about how species identification was conducted, or at least cite the same refs as in the methods.

RESPONSE: We have added this sentence to the Results section: “Prior to generating RADseq data, each wireworm sample was assigned a species designation based on visual morphological traits assessed under a dissecting microscope following the protocols of [22] and [23] (lines 185-187).

1.8) Line 220. While I understand the reference to “within species” analyses/datasets, relative to lines 160-161, it is unclear whether these were used for the within species analyses (3 paragraphs after line 183) or whether the full dataset was used. Perhaps it would be useful to either clarify within these paragraphs or add another sentence around line 160 to clarify what data is being used in what analyses.

RESPONSE: We made the following changes to clarify that we filtered the SNPs independently for each within-species analysis (and therefore we retained different numbers of SNPs for each analysis). (1) We added the following sentence to the Results section: "The number of SNPs retained for analyses conducted separately for each L. californicus subgroup (described further below) ranged from 489 to 1,118, and for each L. infucascatus subgroup ranged from 223 to 330. (lines 195-197) (2) We added/modified the following sentence to the Methods section: "We performed these filtering steps for the full dataset of Limonius samples, and also independently for each known species, as well as for each subgroup within known species. Performing these filtering steps independently for each species and subgroup allowed us to maximize the performance of the within-species and within-subgroup analyses by maximizing the number of SNPs retained, minimizing the amount of missing data, and ensuring singletons and monomorphic SNPs were removed. Notably, performing these steps independently for each data subset resulted in different numbers of SNPs across analyses, as well as slight differences in sample sizes across analyses due to different numbers of individuals passing the missing data filters." (now lines 750-753)

1.9) Lines 318-327. I am not too big a fan of using a straight divergence rate to estimate ages, even if the age estimates are not a large part of this study. I'd be curious what the estimated divergence ages would be using a molecular dating approach, even just with secondary calibrations based on recent phylogenomics studies (e.g. 10.1073/pnas.1909655116) or some of the other older studies based on a few more comparable genes. I'm not saying that this is 100% necessary for this study, it might be worth exploring if other reviewers make similar comments.

RESPONSE: We now include a Bayesian molecular dating analysis using BEAST2, and we report 95% HPD intervals for the divergence dates (lines 371-374, 831-839, and Fig. S8).

1.10) Lines 464-466. This is an important point, and I think could be expanded on. Was/is it possible to estimate the larval instar of these samples, and from that infer age (even as broadly as > or < 5 years old)? Given the long development time, it's possible that all sampled individuals were greater in age than any of the pesticide treatments (max 6 years, if I am interpreting everything correctly)! Alternatively, is the history of these plots known from before the current pesticide application period?

RESPONSE: We added the following sentences: "Although estimating the exact age of the L. canus was not possible, all of the collected samples were determined to be later larval instars based on body length. Since the larval state of L. canus lasts for multiple years [45], the wireworms were likely exposed to insecticides for extended periods. Therefore, it is conceivable that the mechanism driving the genetic signature observed here is high mortality in earlier life stages of larvae maladapted to broad spectrum pesticides." (lines 611-615)

1.11) I wonder also about the genomic locations of these ~250 SNPs. Of course, with 115k contigs, there's going to be a limit to what you can say about genomic location. But if most of these occur on a handful of contigs, that might be something to report. Now that I see the final supplementary table, it doesn't look like any meaningful pattern there, but maybe that's a point to bring up as well (i.e. "no indication of a genomic island of differentiation" or something).

RESPONSE: We have added this sentence to the Results section: “These 244 outlier SNPs were distributed across 163 scaffolds of the reference genome, with no indication of genomic islands of differentiation (Table S3, Table S4).” We have also added Table S4, which provides a summary of all the scaffolds for which more than one outlier SNP occurred; in every case, the outlier SNPs on the scaffold were associated with the same restriction cut site, indicating no large-scale genomic islands of differentiation across scaffolds.

1.12) Line 577. What kind of parameters were used for GATK? Were the “best practices” followed for species without existing genotype data?

RESPONSE: We added that GATK was run “using default parameters” (lines 740-741). The exact commands can also be found on our github page, as referenced in the Statistics and Reproducibility section. We generally followed the GATK best practices, but the details of these best practices are constantly evolving, and therefore we feel that it could be ambiguous to state in the manuscript that we followed GATK best practices. From the GATK web documentation: “Remember that as our work continues and our Best Practices recommendations evolve, specific command lines, argument values and even tool choices described in the paper become obsolete.”
<https://gatkforums.broadinstitute.org/gatk/discussion/11027/how-should-i-cite-gatk-in-my-own-publications>

1.13) Lines 629-630. Was the full HTStream pipeline used here, or just particular parts of it? Are any parameters/settings needed?

RESPONSE: HTStream provides multiple programs with a variety of functions, and does not have one specific pipeline, and so we have reported each HTStream program we used. We have now added a more thorough description of our adapter trimming: “hts_AdapterTrimmer in HTStream (<https://ibest.github.io/HTStream>) was used to trim Illumina adapters from raw RADseq reads and to remove any remaining reads <50bp long” (lines 732-733). The HTStream commands we used can be found on our github page, as referenced in the Statistics and Reproducibility section.

1.14) Line 635. Make reference to the github page containing Python scripts?

RESPONSE: We added a reference to the github page to the end of this sentence (line 809).

1.15) Table 1. Were samples at a particular locality all collected at the same time? Or were localities repeatedly sampled over the 4 year period? I suppose this point only really affects the interpretation of the *L. canus* samples (see comment line 464-466), so maybe it would be sufficient to just specify what year those were collected, either here or in the text (around line 506).

*RESPONSE: We clarified that all *L. canus* samples were collected in 2014 and 2015 (now line 669).*

Reviewer #2 (Remarks to the Author):

Andrews et al. present some nice work identifying cryptic species of wireworm and attempting to find outlier SNPs associated with pesticide treatments. I agree with the authors that genomics has been underutilized in many species of agricultural concern and this paper does a nice job of jumping into a system that had no previous genomic resources. I found the paper to be very well written. Further, the authors provide clear descriptions of what they did in terms of data generation, analyses, and their interpretation of results. All data and code appears to be available. I have a few suggestions that I think would improve the manuscript.

RESPONSE: We are pleased that the reviewer found our work to be a novel and important contribution.

Major (in no particular order):

2.1) I found the mapping results, PCA results, and mtDNA tree to be somewhat at odds. Lines 156-158 say that the lowest average mapping rate was for the *L. infuscatus* samples. This result suggests that this species is very sequence divergent from the reference and the authors point this out (lines 162-165, and more on that below). However, PC1 of Figure 2 (50% of variation) splits *californicus/infuscatus* and *canus*, not *californicus/canus* and *infuscatus* as I would expect. Is the low mapping rate not due to sequence divergence? Further, the mtDNA tree (Figure 7) suggests that *canus* and *californicus* are more closely related, consistent with the mapping results, not the Figure 2 PCA.

RESPONSE: The apparently discrepant results of the PCA are caused by uneven sample sizes across the three species (the influence of uneven sample sizes on PCA is described in McVean 2009, A Genealogical Interpretation of Principal Components Analysis, Plos Genetics). In particular, the larger sample size of L. canus is causing it to separate on the first PC. We now present PCA results with and without subsampling to equal sample sizes for the three species (now Fig. S1 and Fig. 2). After subsampling, the PCA results show the same pattern as the Fst, sNMF, and phylogenetic results, i.e. L. infuscatus separates from the other two species on the first PC, and the other two species separate on the second PC. This is now described in lines 207-210 and 759-762.

2.2) Continuation from above. Given F_{st} is so high (nearly 1) between *L. californicus*, *L. canus* and *L. infuscatus*, some metric characterizing percent sequence divergence (e.g., D_{xy}) between species needs to be included in the paper. It also adds to the comparison of sequence divergence calculated from the mtDNA. Nucleotide diversity would be nice as well.

RESPONSE: We now include Nei's genetic distance values between L. californicus, L. canus, and L. infuscatus for the RADseq data (lines 221-223 and 768).

2.3) Motivation for the relatedness analysis was unclear in the introduction or results, it only became clear when reading the methods at the end of the paper.

RESPONSE: We have now moved the sentence describing the motivation for the relatedness analysis from the Methods section to the Results section (now lines 311-313): "The identification of close relatives within a dataset can provide insight into fine-scale dispersal patterns, since the identification of close relatives in different geographic locations indicates dispersal by at least one individual."

2.4) Why is *L. infuscatus* considered three groups in figure 4b? It seems like it is just two, E and FG. F and G only seem to be splitting apart a slight bit in the PCA due to some sampling/geography pattern that doesn't show up in sNMF analyses (Figure 3) or in mtDNA analyses.

RESPONSE: We chose to provide as detailed a description of the results as possible. We felt that it was important to report that groups F and G separated on the PCA, even though these two groups did not separate in sNMF or mtDNA analyses. Groups F and G are geographically separated, providing evidence that the weak differences seen on the PCA may be biologically meaningful, e.g. could result from an isolation by distance pattern, which could be tested with more sampling across a wider geographic range.

2.5) Did the authors explore how much of an impact including *infuscatus* had on F_{st} estimates between *canus* and *californicus*. Given the low percentage mapping, it must have heavily biased the SNPs. Asked another way, did the authors compare F_{st} when filtering only *canus* and *californicus*, to the F_{st} value they report that is estimated when filtering all three together.

*RESPONSE: We re-ran the SNP filtering with only *L. canus* and *L. californicus* included, and then calculated pairwise F_{st} . The result ($F_{st}=0.847$, $p<0.0001$) was only slightly higher than when *L. infuscatus* had been included in the SNP filtering ($F_{st}=0.816$, $p<0.00001$), indicating a minimal impact of ascertainment bias on this result.*

2.6) I would suggest the authors consider adding a few sentences about the phylogeographic patterns they see and how it fits with other systems. It seems to me that there are several vertebrate and invertebrate examples that show distinct phylogeographic breaks consistent with the two cryptic species of *L. infuscatus* and the *L. californicus* group A split with the other three groups.

RESPONSE: Done. We now include several sentences describing studies that show similar phylogeographic patterns to those we found for wireworm species. Lines 493-497: "Molecular dating placed the divergence times for the cryptic species in the Pleistocene. Many other animal and plant species within this geographic region have phylogeographic structure that dates to this time period, and this structure is thought to be driven by reproductive isolation in Pleistocene glacial refugia, followed by range expansions after glaciers retreated [36-38]."

The new references we cite include a paper about beetle phylogeography (Maroja et al. 2007) and two papers summarizing phylogeographic patterns for a wide taxonomic range of species (Brunsfield et al. 2001, Rankin et al. 2019).

2.7) Were sex linked SNPs or scaffolds used in the analysis? Do you know the sex of any of the samples?

If so, do they correspond to any of the outliers in PCAs or to the F and G groups?

RESPONSE: Unfortunately, the reference genome has not yet been annotated, and therefore we are unable to determine whether sex-linked SNPs are present in our data. In addition, we do not know the sex of any of the samples, since during the larval stage there are no known morphological differences between the sexes. We agree this would be an interesting question to pursue after the genome has been annotated.

2.8) I was wondering if the authors could comment on if their outlier SNPs are near, or in, genes with functional relevance with respect to pesticides? Is their reference genome annotated?

RESPONSE: Unfortunately, the reference genome has not yet been annotated. We clarify this in lines 627-628.

2.9) There seems to be some inconsistency between plots and tables that needs explaining/fixing. For *infuscatus*, Table 2 says there are 3 from group F and 7 from group G (10 total). It also says this in the text (Line 211). However, Figure 4 b shows 1 dot for F + G. Further, figure 3 seems to show 12 bars for F + G. I didn't double check all the others.

RESPONSE: Because we conducted SNP filtering independently for each data subset, we ended up with slight differences in sample sizes across analyses, since the filtering includes a step to remove samples with certain percentages of missing data. We clarify this by adding the following sentence to the Methods section: "Notably, performing these [filtering] steps independently for each data subset resulted in different numbers of SNPs across analyses, as well as slight differences in sample sizes across analyses due to different numbers of individuals passing the missing data filters." (lines 750-753). Note that all analyses conducted for a given group or subgroup use the same SNPs. The total number of samples for group F+G is 12 for both the PCA (Fig. 4b) and the sNMF (Fig 3). The PCA appears to have 11 dots because two dots are very close together.

Minor:

2.10) Figure 7. *L. canus* should be labeled. I would also suggest adding the species name by the groups in the main part of the figure.

RESPONSE: Done. We added labels for L. canus, L. infuscatus, and L. californicus to the tree (now Fig. 6).

2.11) Lines 144-145. A few commas are missing for numbers.

RESPONSE: Done.

2.12) Size of the scaffolds for the Fst outlier snps would be useful (table S2). Some of these could be in close proximity to one another, especially if they are all located on small scaffolds.

RESPONSE: We have now added scaffold lengths to the table (which is now Table S3), and we also now include Table S4, which provides a summary of the genetic distances between outlier SNPs occurring on the same scaffold (see comment above).

2.13) Conclusions statement might need to be more nuanced. I don't believe you conclusively 'identified genomic adaptation to pesticide treatment'.

RESPONSE: We changed the wording to state that we identified "putative" adaptation: "identified... putative genomic adaptation to pesticide treatment." (now lines 634-5).

2.14) Figure 4. Consider including species name in corner of each a) b) c) plot. Makes it easier for the reader to remember what species they are looking at without jumping back to text.

RESPONSE: Done.

Reviewer #3 (Remarks to the Author):

Overall this is a fantastic study that should be accepted for publication. The in-depth study of the different *Limonium* populations and species is a great resource for future IPM efforts in the region, as well as future population studies for these species. These data are presented clearly, and the figures are very informative. The methodology is also presented clearly, allowing for reproduction of these finds, as well as applications to other wireworm species.

RESPONSE: We are encouraged by the reviewer's comment about the quality and presentation of our work.

3.1) One minor typographical issue is a missing period on page 15, line 336. I did not notice any others, but a good recommendation is another skim through by the authors before final publication.

Response: Done.

3.2) The only issue that I would like to bring up here is more philosophical, rather than a rejection of the conclusions drawn in this work. I would like all of the author's to really consider this alternative interpretation. The DNA data presented is all mitochondrial DNA, yet frequently the authors mention that different populations do not interbreed (pg 16), and also mention one individual as a possible hybrid (pg 17). However, the null hypothesis of mitochondrial DNA inheritance is uniparental inheritance from the mother only. Exceptions to this rule exist, but have not been demonstrated within elaterids (i.e. wireworms). Looking at mitochondrial DNA would not show interbreeding between populations, as only the mother would pass these genes down. There would also be no hybridization or recombination events, as the mitochondrial genome is a single chromosome inherited from a single parent. I argue that these data actually demonstrate the poor dispersal by female *Limonium*. While these data suggest deep divergences between populations, potentially leading to novel traits in

each population, we can make no claim about the lack of interbreeding here, as these events would not be seen if only males travel long distances as their mito-genomes would not appear. As most of the references included cite male dispersal, the authors need to consider this possibility. The presence of two nearby female lines of *Limonius infuscatus* within a field argue, in my opinion, that interbreeding is possible

Now, in my personal observations, there are some morphological variations between these populations, suggesting that you are all correct that there may be cryptic species present within *Limonius californicus* and *L. infuscatus*. The half-million to million year divergence time in the mito-genome groups is also indicative of long periods of isolation. However, we can't say for certain that these may be cryptic species yet. It may be that while females disperse poorly, allowing for isolated mito-genome lines in the valleys, but males could be stronger dispersers. The males may move nuclear genes between populations, keeping them all under the umbrella of a single species, despite distinct and isolated female mitochondrial populations. These also may be relicts of glaciation trends at the end of the last ice-age on the populations.

In short, I would like the authors to acknowledge the limitations of interpreting mitochondrial DNA on the possibility of interbreeding between populations. No claims of this can be made due to the nature of mitochondrial inheritance. This actually stresses the need to examine nuclear genes to evaluate the possibility of cryptic species or if these populations interbreed. I look forward to reading that paper when you all get around to that!

The above points have no effect on the validity of publication. The authors are free to reject the above points, but I would like them all to think on them. I think their arguments would be strengthened if they mention them in their discussion of their data .

Response: It is very interesting to hear that the reviewer has observed morphological differences between the putative cryptic species. We agree that with mitochondrial DNA alone, we would not be able to make strong conclusions about hybridization or reproductive isolation of males. However, the reviewer misunderstands that our results are not based on mitochondrial data alone. The RADseq data in our study include thousands of nuclear markers from across the genome; we now clarify this in lines 154-5 (now reads "(RADseq), a technique that surveys thousands of loci across the nuclear genome"). Because these nuclear markers show the same patterns of genetic structure as the mtDNA sequence data, we believe that the conclusions in the manuscript about reproductive isolation and hybridization are valid.

REVIEWERS' COMMENTS:

Reviewer #1 (Remarks to the Author):

The authors have done a fine job addressing all of my original comments and those of the other reviewers. My only comment is extremely minor: I'd suggest using "larval stage" instead of "larval state" at line 435.

Reviewer #2 (Remarks to the Author):

Andrews et al. have done a very thorough job of addressing all reviewer comments and I am excited to see this published. I have no further suggestions and look forward to more from this system.

Reviewer #1 (Remarks to the Author):

The authors have done a fine job addressing all of my original comments and those of the other reviewers. My only comment is extremely minor: I'd suggest using "larval stage" instead of "larval state" at line 435.

Response: We are happy to receive this positive review. We changed "larval state" to "larval stage" (line 435).

Reviewer #2 (Remarks to the Author):

Andrews et al. have done a very thorough job of addressing all reviewer comments and I am excited to see this published. I have no further suggestions and look forward to more from this system.

Response: We are pleased that the reviewer was satisfied with our revisions.